



# Water and sediment fluxes in Mediterranean mountainous regions: Comprehensive dataset for hydro-sedimentological analyses and modelling in a mesoscale catchment (River Isábena, NE Spain)

Till Francke[1], Saskia Foerster[2], Arlena Brosinsky[1,2], Erik Sommerer[7], Jose A. Lopez-Tarazon[1,3,4], Andreas Güntner[7,1], Ramon J. Batalla[4,5,6], Axel Bronstert[1]

[1] University of Potsdam, Institute of Earth and Environmental Science, Karl-Liebknecht-Str. 24-25, 14476 Potsdam, Germany
[2] GFZ German Research Centre for Geosciences, Section 1.4 Remote Sensing, Telegrafenberg, 14473 Potsdam, Germany
[3] Mediterranean Ecogeomorphological and Hydrological Connectivity Research Team (MEDhyCON), Department of Geography, University of the Balearic Islands, 07122 Palma, Spain
[4] RIUS, Fluvial Dynamics Research Group, University of Lleida, E-25198 Lleida, Catalonia, Spain
[5] Catalan Institute for Water Research, E-17003, Girona, Catalonia, Spain
[6] Faculty of Forest Sciences and Natural Resources, Universidad Austral de Chile, Valdivia, Chile

[7] GFZ German Research Centre for Geosciences, Section 5.4 Hydrology, Telegrafenberg, 14473 Potsdam, Germany

*Correspondence to*: T. Francke (francke@uni-potsdam.de)

**Abstract**

A comprehensive hydro-sedimentological dataset for the Isábena catchment, NE Spain, for the period 2010-2016 is presented to analyse water and sediment fluxes in a Mediterranean meso-scale catchment. The dataset includes rainfall data from twelve rain gauges distributed within the study area complemented by meteorological data of twelve official meteo-stations. It comprises discharge data derived from water stage measurements as well as suspended sediment concentrations (*SSC*) at six gauging stations of the River Isábena and its sub-catchments. Soil spectroscopic data from 351 suspended sediment samples and 152 soil samples were collected to characterize sediment source regions and sediment properties via fingerprinting analyses. The Isábena catchment (445 km²) is located in the Southern Central Pyrenees ranging from 450 m to 2,720 m a.s.l., together with a pronounced topography this leads to distinct temperature and precipitation gradients. The River Isábena shows marked discharge variations and high sediment yields causing severe siltation problems in the downstream Barasona Reservoir. Main sediment source are badland areas located on Eocene marls that are well connected to the river network. The dataset features a wide set of parameters in a high spatial and temporal resolution suitable for advanced process understanding of water and sediment fluxes, their origin and connectivity, sediment budgeting and for evaluating and further developing hydro-sedimentological models in Mediterranean meso-scale mountainous catchments.





The dataset is available at http://doi.org/10.5880/fidgeo.2017.003 .

**Keywords.** Rainfall, discharge, suspended sediment concentration, soil spectroscopy, fingerprint properties, meso-scale

## 1    Introduction

Many dryland regions experience strong erosion in headwater catchments, resulting in reservoir sedimentation and a loss of storage volume that can lead to a significant reduction of water availability within few decades. This calls for a thorough analysis of water and sediment fluxes in regions affected by high erosion and sediment delivery rates as a prerequisite for sustainable water and sediment management. While specific processes of the hydro-sedimentological system, e.g. hillslope water dynamics, plot-scale erosion, fluvial morphology of river stretches, sedimentation phenomena in lakes and reservoirs, have been studied in detail, there has been much less research on process analyses including the linkages and interactions between landscape compartments on catchment level that are relevant for water and sediment management. Management decisions should ideally be based on in-depth analyses and modelling with the aim to improve understanding of water and sediment fluxes, connectivity of landscape compartments as well as origins of sediments. For this purpose, long-term consistent multi-scale measurements are needed. There is, however, a lack of comprehensive high-resolution, multi-year hydro-sedimentological datasets due to the enormous effort to run measurement programmes, particularly in mountainous regions. Furthermore, publication of comparable datasets has been done rather dilatory in the past. Notable exceptions include the datasets by Goodrich et al. (2008); Nichols et al. (2008) and Stone et al. (2008) (hydro-meteo-sedimentological data of small semiarid catchments), Nord et al. (2016) (hydro-meteorological data of a mesoscale Mediterranean catchment in S-France) and Beaulieu (2016) (glaciated catchments, NW-USA).

With this publication, we present a comprehensive hydro-sedimentological dataset for the Isábena catchment, NE Spain, spanning the period 2010-2016 to support the analysis and modelling of water and sediment fluxes in highly erodible mountainous dryland regions. The Isábena catchment in the Southern Central Pyrenees is an ideal observation site, since it features very high erosion and sediment delivery rates, large temporary sediment storage and re-mobilisation in the river system and considerable effects of partially (dis-)connected sediment fluxes. The catchment shows a notable heterogeneity in terms of land use and lithology and strong orographic and pluviographic gradients.

The specificity of this dataset is the long measurement period of more than six years including several high discharge events, the comparatively large size of the catchment covered in a multi-site measurement programme, and the extensive set of measured parameters. The dataset consists of meteorological data (rainfall, air temperature, solar radiation, air humidity),



hydrological data (river water stage, discharge), sediment data (suspended sediment concentration) as well as spectral reflectance data of samples of suspended sediment and soil source areas. Additionally, for two adjacent sub-catchments (~70 km²) of the River Isábena, namely Villacarli and Carrasquero, an airborne hyperspectral image dataset acquired in April and August 2011 with simultaneous comprehensive ground-truth data collection as well as an airborne LiDAR dataset of the

same area acquired during the August 2011 campaign are available in a separate data publication (Foerster et al., 2015).

The comprehensive hydro-sedimentological dataset is an outcome of a long-lasting and on-going joint research cooperation between the University of Potsdam and the German Research Centre for Geosciences (GFZ), Germany, together with the University of Lleida and the Forest Sciences Centre of Catalonia, Spain. The research was mainly conducted within the

projects "SESAM: Sediment Export from large Semi-Arid Catchments: Measurement and Modelling" (2004-2008) and "Generation, transport and retention of water and suspended sediments in large dryland catchments: Monitoring and integrated modelling of fluxes and connectivity phenomena" (2010-2014), both funded by the Deutsche Forschungsgemeinschaft (DFG). The focus in the first project was on studying soil erosion, sediment transport and reservoir sedimentation by setting up a continuous monitoring programme and developing a meso-scale process-based model. The

latter project mainly focused on studying the interrelation of transfer, storage, re-entrainment as well as connectivity processes through a comprehensive hydro-sedimentological and spectral measurement programme at plot, hillslope and river scale and remote sensing data analysis as well as through advancing the integrated model for water and sediment fluxes to account for connectivity and scaling issues. The highly dynamic water and sediment fluxes in the Isábena catchment have been analysed in detail based on (parts of) the dataset published here (e.g. Francke et al., 2014; López-Tarazón et al., 2012;

López-Tarazón and Batalla, 2014). Sediment connectivity for two sub-catchments of the Isábena basin was analysed based on digital elevation models obtained from airborne LiDAR data and surface cover fractions obtained from airborne hyperspectral imagery (Foerster et al., 2014), while sediment origins were traced back by spectral fingerprinting based on measured spectra of suspended sediment samples and soil samples of potential sediment sources (Brosinsky et al., 2014a, 2014b). Furthermore, for the process-based, spatially semi-distributed model WASA-SED developed for water and sediment

transport in dryland catchments (Mueller et al., 2010; Bronstert et al., 2014), a framework for studying the effect of model enhancements by, e.g., a higher number of meteorological monitoring stations, improved water and sediment calibration data, and modifications in process formulations, was tested in the Isábena study area based on parts of the dataset presented here.



## 2   Study area

The Isábena catchment, which drains an area of 445 km$^2$, is located in the Southern Central Pyrenees (NE Iberian Peninsula) (Fig. 1). It is the main tributary of the River Ésera; both rivers are the most important tributaries of the River Cinca, in turn the second largest tributary of the Ebro (Fig. 1). The Isábena catchment is characterized by a heterogeneous relief and

pronounced topography (altitude ranging from 450 m to 2,720 m a.s.l.), leading to marked temperature and precipitation gradients. The catchment has a wet and cold Continental Mediterranean climate, with both Atlantic and Mediterranean influences (García-Ruiz et al., 2001). Mean annual precipitation ranges from 450 mm in the lowlands to 1,600 mm in the upland areas, with basin-average annual precipitation of 770 mm.

Flow regime in the River Isábena are characterized by a pluvial regime, with contributions from snowmelt from the upper parts of the catchment. Floods typically occur in spring, triggered by frontal precipitation events, sometimes in combination with snowmelt, and in late summer and autumn caused by localized thunderstorms. The mean annual discharge estimated at the outlet of the basin (Capella gauging station, EA047; Fig. 1) is 4.1 m$^3$ s$^{-1}$, but maximum discharges of up to 370 m$^3$ s$^{-1}$ have been observed (August 1963, return period of 94 years calculated by the Gumbel method from a series of annual

maximum instantaneous discharges for the period 1945-2015); minimum discharges are below 1 m$^3$ s$^{-1}$, but the river never dries up. The mean annual water yield is 177 hm$^3$, a value that represents ~1.5 % of the total runoff in the whole Ebro river basin (López-Tarazón et al., 2009).

The catchment is not hydraulically regulated, thus its hydrological regime is determined by natural factors only. However, the Isábena drains into the Barasona Reservoir, being responsible (together with the Ésera) for its severe siltation due to the

large amounts of suspended sediments which both rivers deliver. Instantaneous suspended sediment concentrations of up to 350 g l$^{-1}$ have been measured at the basin outlet (López-Tarazón et al., 2009), generating mean suspended sediment loads above 250,000 t y$^{-1}$ (i.e. period 2005-2010; López-Tarazón and Batalla, 2014) ) which correspond to a specific yield of about 600 t km$^2$ y$^{-1}$. This can be considered high to very high in comparison with catchments of the same size in this and other regions (Francke et al., 2014; López-Tarazón et al., 2012; de Vente et al., 2006) Sediments mainly originate from the central

part of the catchment (Fig. 1), in a corridor of Eocene marls with sandstones. The marls are exposed to the surface in badland structures (bare surfaces on highly erodible sediments), being the main sediment sources of the catchment (López-Tarazón et al., 2009) despite their small area (less than 1 % of the Isábena catchment area).






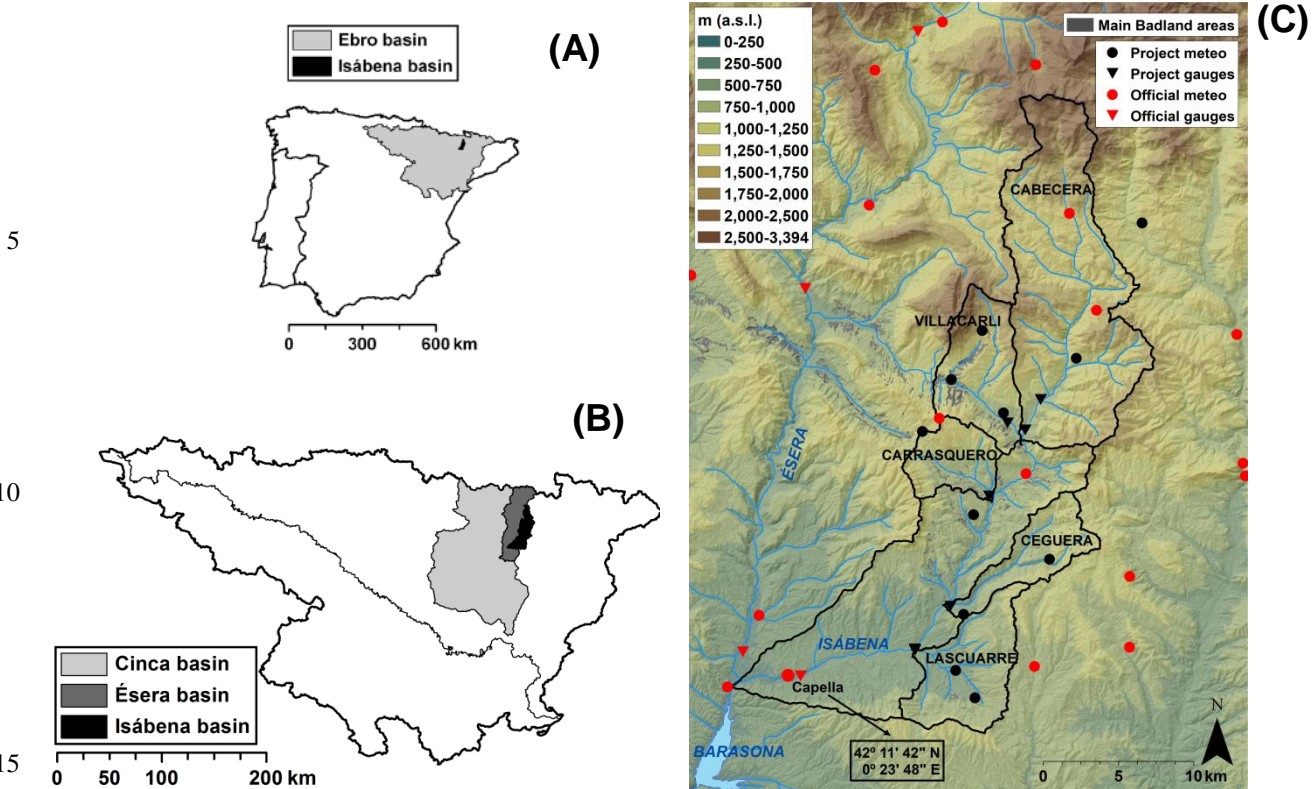

**Figure 1: A) The Isábena in the Ebro basin and in the Iberian Peninsula. B) Location of the Isábena, the Ésera and the Cinca catchments within the Ebro river basin. C) DEM of the Isábena basin, where the location of the main badland areas and the Barasona Reservoir have been pointed out; all the different monitoring stations, both the official ones and those which were installed during the duration of the project have been highlighted.**

## 3 Methods

Water and sediment fluxes have been monitored in the Isábena catchment in the context of hydro-sedimentological research since 2004. Additionally, the Ebro Water Authorities (CHE) maintain a network of monitoring stations for precipitation, temperature, radiation and discharge for operational purposes (*Sistema Automático de Información Hidrológica*, SAIH) of which some of the data are included in the presented dataset. Beside these hydro-meteorological time series, the dataset also contains data on spectral properties of soil and sediment samples from terrestrial and riverine parts of the catchment (see Table 1 for an overview). The details of the dataset (measurement techniques, data processing and checking) are given in the following sections. All data are provided according to the CUAHSI HIS standard, which is explained in section 4. Table 1 provides an overview of the content of the database.





Other data acquired during the research activities are not part of this publication. They include multiple sets of space and airborne imagery, terrestrial laser scans of river sections and of badland sites, water and sediment time series of three gauging stations upstream gauge Villacarli, and data on particle movement and soil moisture in a badland site. These data can be requested from the authors, where processed and licensing permits it.

5   **Table 1: Summary of data contained in dataset**

| Variable | No. of stations / sites | Instrument / Source | Resolution [min] | Variable Code in database |
|---|---|---|---|---|
| Precipitation | 18 | tipping bucket | breakpoint, 1, 15 | rainfall_* |
| Air temperature | 9 |  |  | temperature |
| Solar radiation | 2 | official stations (SAIH) | 15 | global_radiation |
| Air humidity | 2 |  |  | relative_humidity |
| Water stage | 5 | Capacitive, micro-wave, shaft-encoder | 1, 5, 15 | water_stage_* |
| Discharge | 5 | dilution, current meters | intermittent | discharge_metering discharge_* |
|  |  | from water stage | 1, 5, 15 |  |
| Turbidity | 3 | turbidimeters | 1, 5, 15 | turbidity_* |
| Suspended sediment concentration | 5 | manual and automatic samplers | Intermittent, event-based | ssc_sampled |
|  | 3 | from turbidity | 1, 5, 15 | ssc_continuous_* |
| Spectra of soil samples | 152 | In-situ and laboratory spectroscopy | - | wavelength_nm_350-2151 |
| Spectra of sediment samples | 351 |  | - | wavelength_nm_350-2151 |

"*" denotes further suffixes in the names indicating the respective resolution, e.g. "rainfall_15"

## 3.1   Meteorological data

Within the project context, rainfall was the only meteorological variable monitored at 12 stations (3.1.1). However, other
10   meteorological variables (e.g. temperature, radiation) were measured and operationally collected by the SAIH network
(3.1.2).



### 3.1.1    Rainfall

A total of 12 rain-gauges were installed and distributed across the catchment and vicinities to complement the official coverage of the rainfall network (18 official rain gauges are located within and/or close to the Isábena catchment) (Figure 1). Two different types of rain gauges were installed: 1) a tipping bucket precipitation transmitter 5.4032.XXX (Thies Clima, Göttingen, Germany) connected to a data logger CR-1000 (Campbell Scientific Ltd., Shepshed, England) was installed in Villacarli and in Lascuarre outlets (Figure 1), while 2) a Davis rain gauge connected to a Hobo Event data logger was installed in the rest of the project rainfall stations. In the case of the official precipitation stations (Figure 1) all the installed rain gauges were different models of tipping bucket transmitter (Thies Clima, Göttingen, Germany). All rainfall data have been checked and periods of malfunctioning (i.e. clogged funnels) been removed by comparing plots of cumulative rainfall sums of adjacent stations.

### 3.1.2    Air temperature and humidity, Solar radiation

These meteorological variables have been courteously provided by the Ebro Water Authorities. They are operationally monitored within the SAIH network (Automatic System of Hydrological Information). Thus, postprocessing was limited to the removal of periods of implausible data and visual checks. While we chose not to alter the data, some observations may be of interest to the potential user: Global radiation time series show a slight but steady decrease in recorded values over time. A similar behaviour is present in the time series of relative air humidity at the station Barasona. Likewise, recorded maximum values of relative air humidity decrease slightly but steady over time. At El Grado (5 km W of Barasona Reservoir), recorded values of relative air humidity jump to higher values in 2012. In both time series of relative air humidity values never reach 100 %.

## 3.2    Hydrological data

Streamflow was monitored at the outlet of the catchment and of five sub-catchments by continuously measuring water stage (3.2.1). Occasional discharge metering (3.2.2) allowed for defining the rating curves for the conversion of water stage into time series of discharge (3.2.3).

### 3.2.1    Water stage

Water stage was monitored at the outlet of the catchment and of five sub-catchments (Figure 1). Capella is the official gauging station of the Isábena catchment. Its broad-crested weir forms an artificial cross section and is equipped with a shaft encoder for water stage measurements, recorded every 15 minutes. For the sub-catchments, water levels were measured at



cross sections close to the sub-basin outlets where access and conditions in terms of comparatively well defined and stable cross sections were favourable (i.e. bridges). Table 2 summarizes the specifications of each gauging station. At Villacarli, a microwave stage recorder (RQ 24; Sommer GmbH, Koblach, Austria) was used to measure water level and flow velocity in high temporal resolution of 1 min. At all other stations, capacitive water-stage loggers (WT-HR; TruTrack Ltd.,

Christchurch, New Zealand) with a temporal resolution of 5 min were used. Manual readings were performed regularly to verify and complement the automatic data. The water stage records were calibrated against these readings at a fixed gauge datum to account for sensor drift.

For gauges Carrasquero, Ceguera and Villacarli (see Figure 1), changes in the geometry of the streambed were observed after heavy floods. The time of these changes have been reconstructed manually; the respective difference to the effective

flow level has been added as offset information in the database. Thus, both the absolute water level $h_{abs}$ and the effective water level $h_{eff}$ can be retrieved with

$$h_{eff} = h_{abs} - \text{offset}$$

For the subsequent steps, $h_{eff}$ was used.

**Table 2: Summary of stream gauge properties**

| Gauge | Setting | Stage recording | Resolution [min] | Number of meterings |
|---|---|---|---|---|
| Cabecera | broad-crested weir | capacitive | 5 | 18 |
| Capella | broad-crested weir | shaft encoder | 15 | 6 |
| Carrasquero | gravel-bed below bridge | capacitive | 5 | 10 |
| Ceguera | gravel-bed below bridge | capacitive | 5 | 13 |
| Lascuarre | corrugated metal pipe | capacitive | 5 | 9 |
| Villacarli | gravel-bed below bridge | micro-wave | 1 | 30 |

### 3.2.2 Discharge metering

At all stream gauges, repeated manual discharge meterings with the velocity-area method or the dilution method were conducted to establish and extend the gauge-specific water stage - discharge relationships (rating curves) (Table 2). One of

the two methods was chosen according to the actual site and flow conditions at the time of the measurement. At low flow conditions and when the stream cross section was not inaccessible due to too high flow, the salt dilution method was used with tracer equipment (TQ-Tracer, Sommer GmbH, Koblach, Austria) or conductance meters (Cond 3310; WTW GmbH, Weilheim Germany). During all other conditions, current meters (C2; Ott GmbH, Kempten, Germany) and acoustic devices



(Ott-ADC; Ott GmbH, Kempten, Germany; FlowTracker, SonTek, San Diego, USA) were used. The respective metering data are included in the database. Associated measurement uncertainties were estimated based on recommendations from multiple sources (Le Coz et al., 2014; Pelletier, 1988; Westerberg et al., 2011).

### 3.2.3 Discharge time series

Table 2 lists the number of discharge meterings available for each river gauge. For the catchment outlet, water level is converted into discharge by means of the water stage – discharge rating curve developed by the authors (for more information, see López-Tarazón et al. 2010), which included the effect of the high suspended sediment concentrations that usually flow through the cross-section (hence likely decreasing turbulence, thus providing a higher discharge at the same water level in comparison to clean water). For the sub-catchments, these meterings were used to construct rating curves

using the Bayesian curve fitting (BaratinAGE, vers. 2.1, Le Coz et al., 2014) and to derive the time series of discharge from the water stage series (see 3.2.1). This procedure additionally yielded the associated 95%-confidence interval CI (for water stage and the parametric uncertainty). As the CUAHSI standard only supports recording symmetrical uncertainty measures, CI was approximated as symmetrical and included in the data table in the CUAHSI field "ValueAccuracy" as 0.5 * CI.

As some of the instruments yielded a "stepped" time series of stage, the resulting discharge time series were step-corrected

using the RHydro-package (Reusser et al., 2012), which removed plateaux by interpolating the respective adjacent non-plateaux values.

## 3.3 Suspended sediment concentration

### 3.3.1 Water samples

At all stream gauges, water samples were taken manually and automatically with automatic samplers (ISCO 3700) on an

event-based scheme. These samples were then filtered or decanted for higher concentrations. The resulting sediment was oven-dried and weighted ($m_{Sed}$, g), to allow the computation of initial suspended sediment concentration $SSC$ (g/l) as

$$SSC = m_{Sed} / V_{Water}$$

where $V_{Water}$ is the volume of water abstracted from the sample (l). The respective $SSC$ data is part of the database.

The representativity of side-samples in relation to average $SSC$ at the section can generally be assumed to be acceptable due

to the observed high flow turbulence and the predominantly fine sediment fractions.



### 3.3.2    Turbidity measurements, continuous *SSC*

At the gauges Villacarli, Lascuarre and Capella, turbidimeters recorded turbidity in 5, 5 and 15 min (from averages of 5 s instrument readings), respectively. The installed turbidimeters are Turbimax WCUS41 Endress+Hauser AG, Reinach, CH (Capella, until Nov 2015) and WTW ViSolid (all other periods). The turbidimeters are then connected to Campbell CR-510 / CR-1000 (Campbell Scientific Inc., Logan, Utah, USA) or MIQ (WTW) data loggers which record and store the raw turbidity values. The database contains these raw data of the sensors (in mA, g $SiO_2$ and turbidity units, respectively).

Despite the autocleaning feature of the sensors, sensor signal showed more (Villacarli) or less (Lascuarre, Capella) pronounced deterioration over time, namely by an increasing zero-level with occasional sudden drops. This offset has been corrected manually and stored along in the database ($turbidity_{eff}$ = $turbitidy_{raw}$ − offset). For some periods, the sensor signal appeared completely corrupted (e.g. due to sensor burial, extreme low flow). These periods have been tagged in the database via the QualityControlLevelCode-field as erroneous.

Using the abovementioned lab-samples, the valid turbidimeter data was converted to *SSC* using quadratic regression, constituting separate time series in the database.

## 3.4    Spectral data

Spectral analyses were carried out for soil and sediment samples taken from the study area. This comprises 351 sediment samples from the river collected during flood events (see section 3.3.1) and 152 soil samples from the main potential sediment source areas within the Isábena catchment. Details on sampling and spectral measurement procedures can be found in the subsequent sections. Further details on the spatial location of sampling sites as well as the use of these data in (spectral) fingerprinting studies to trace back the origin of suspended sediments are given in Brosinsky et al. (2014a, 2014b).

### 3.4.1    Spectra of soil samples from source areas

Source area soil samples were collected during two field campaigns in October 2010 and June 2011. Sampling sites were selected based on previous analyses of land use distribution (MARM, 2008), erosion susceptibility (Fargas et al., 1997) and accessibility. This sampling included all major land use types, namely forest (covering 47 % of the catchment area), shrubland (30 %), agricultural land (14 %), and grassland (8 %). In addition, potential source areas that cover only small parts of the river basin (< 1 %) but that were suspected to contribute high proportions of suspended sediment were sampled, including badlands, unpaved roads and open slopes exposing soil next to roads or channels. Sampling sites were chosen in close vicinity (< 100 m) to stream or river reaches to ensure the connectivity of potential sources to the river network. At



each site, five grab samples of easily erodible material (top 1−3 cm) were collected from a representative area of approximately 5 m × 5 m in size.

Spectral reflectance was measured as the ratio of reflected radiation to the total radiation incident on a surface as a function of wavelength (Baumgardner et al., 1985), both 1) in situ during material collection, and 2) in the laboratory following some

preprocessing of the samples. In both cases, an ASD FieldSpec3 High-Res portable spectroradiometer (Analytical Spectral Device Inc., Boulder, USA) was deployed, using a white reference (95 % Zenith Alucore Reflectance Target, SphereOptics GmbH, Uhldingen, Germany) as standard. The ASD spectroradiometer acquires 2,151 channels in the 0.35−2.5 μm spectral range at a true sampling interval of 1.4 nm in the VNIR region (0.35−1.0 μm) and 2 nm in the SWIR region (1.0−2.5 μm).

In-situ reflectance spectra were collected just before grab sampling at the corresponding location with an accessory light

source (contact probe) mounted on the light-collecting head of the ASD spectroradiometer. This keeps illumination conditions stable and excludes atmospheric influences for all measurements.

For laboratory measurements, the source material collected from the five locations per site was thoroughly mixed to provide homogeneous samples, dry sieved to 63 μm to minimize differences in particle size composition between source and sediment material (e.g. Peart and Walling, 1986; Smith and Blake, 2014), placed in shallow 5 cm × 5 cm plastic containers,

and oven dried at 60 °C for 24 h prior to spectral measurements. Then, spectral readings were taken in a dark room facility using the same ASD spectroradiometer and white reference as in the field. Illumination was provided by a 2,000-W lamp installed at approximately 80 cm from the sample at a zenith angle of 45°, and the optical head of the ASD was mounted perpendicular to the sample. For each sample, four readings were taken and subsequently averaged, with the sample rotated 90° after every reading to reduce illumination effects. For more details see Brosinsky et al. (2014a).

**3.4.2     Spectra of suspended sediment samples**

The sampling of suspended sediment material by means of automatic ISCO samplers is described in section 3.3.1. All spectral reflectance measurements of suspended sediment material were taken in the laboratory similar to those of source material (see 3.4.1).

Four events sampled at the catchment outlet (Capella gauging station) were chosen for spectral analysis, namely the events

of (a) 24th/25th September 2011, (b) 22nd March 2012, (c) 3rd/4th June 2012, and (d) 20th June 2012. For that purpose, the material was gently disaggregated using pestle and mortar, if necessary, and further processed and measured as the source area material in the laboratory (section 3.4.1).

In addition to material from Capella, suspended sediment collected at the five sub-catchment outlets was prepared for analyses. However, *SSC* at some sub-catchment outlets was often substantially lower and thus the amount of collected

material was insufficient for spectral laboratory measurements as described above for which > 2 g are required. Therefore,



these samples were either measured using the sediment layer on the glass fibre filters that remained from *SSC* analyses, or from glass fibre filters specifically prepared for the spectral measurements. Again, spectral measurements followed the laboratory protocol described above. Since we detected spectral differences between loose material and material on filters that are most likely caused by the alignment of sediment particles during vacuum filtration, the measurements should not be compared directly.

For each sample (in-situ, loose source material, loose sediment material and sediment on filters in the lab, respectively), mean reflectance spectra were calculated and detector jumps at 1.0 and 1.83 μm occurring on rare occasions were corrected by adaptation to the first detector. All spectra were then smoothed using a Savitzky–Golay filter (Savitzky and Golay, 1964) with a Kernel size of 7.

## 4    Data availability

To maximize the ease of reusability of the data, we created a database following the CUAHSI HIS standard proposed by the Consortium of Universities for the Advancement of Hydrologic Science (Couch et al., 2014; Horsburgh et al., 2008).

This standard defines a structure of a relational database, which allows the unequivocal storage of measured values and the associated metadata. Besides being listed in a central directory (HIS Central Catalog) for improved visibility, the dataset can be explored using a variety of standalone tools, APIs or packages to access the data, i.e. HydroDesktop (Ames et al., 2012), HydroClient, WaterML R-package (Kadlec et al., 2015).

The database is directly accessible via http://hydroportal.cuahsi.org/isabena/cuahsi_1_1.asmx?WSDL . The necessary infrastructure was courteously provided by the CUAHSI Water Data Center. The dataset is published through GFZ Data Services (Francke et al. (2017), http://doi.org/10.5880/fidgeo.2017.03), which provides the raw text files in CUAHSI HIS standard.

## 5    Conclusions

We present an extensive dataset of meteorological, hydrological and sediment data including time series, individual meterings and spectral reflectance data from multiple monitoring and sampling points throughout the meso-scale River Isábena basin. All time series data consistently cover a monitoring period of seven years (2010-2016), comprising years and seasons with markedly different average meteorological conditions, and several major flood events. The dataset allows studying discharge dynamics in different parts of the catchment and its response to the spatio-temporal characteristics of the driving rainfall fields. It enables linking sediment fluxes from the highly erosive upland parts of the catchment, badland areas

in particular, along the river network to the catchment outlet where severe siltation affect the downstream Barasona Reservoir, and tracing back the suspended sediment collected at the outlets of (sub-)catchments to its source areas. Overall, the data support an advanced understanding of water and sediment fluxes and budgets and related sediment connectivity processes and phenomena in mountainous Mediterranean catchments. In its high spatial and temporal resolution and

comprehensiveness, the dataset is expected to serve as a highly valuable benchmark for evaluating, calibrating and further developing hydro-sedimentological catchment models. It holds further potential in aiding the design of long-term sediment management programmes in highly active geomorphic regions with an important role in water resources sustainability, such as the Isábena.

## 6    Copyright statement

All data are subject to free use under a Creative Commons Attribution 4.0 International (CC BY 4.0) Licence.

## 7    Appendices

The digital annex contains the scripts used in the processing of the data. It also provides detailed comments on single steps of the data processing.

## 8    Author contributions

Till Francke initiated the drafting of the paper, designed parts of the monitoring program, supervised or conducted the revision of meteo, discharge and sediment flux data, and coordinated the generation of the database.

José A. López-Tarazón mediated data flow with SAIH, attended gauge Capella and its data.

Arlena Brosinsky and Saskia Foerster conducted all sampling, processing and analyses of the spectral data.

Erik Sommerer contributed large parts of the station maintenance, specifically at gauge Villacarli.

Andreas Güntner contributed in developing the monitoring design and installation of monitoring stations.

All of the abovementioned scientists actively participated in the fieldwork campaigns and contributed sections of the paper.

Ramon Batalla, Andreas Güntner, Saskia Foerster and Axel Bronstert acquired project funds, supervised the project and revised the final version of the manuscript.



## 9 Acknowledgements

The collection and processing of the data presented in this paper were mainly funded within two research projects by the Deutsche Forschungsgemeinschaft (DFG): SESAM ("Sediment Export from Large Semi-Arid Catchments: Measurements and Modelling", 2004-2009, BR 1731/3) and WASESAC ("Generation, transport and retention of water and suspended sediments in large dryland catchments: Monitoring and integrated modelling of fluxes and connectivity phenomena", 2010-2015, BR 1731/4-1-2). Data collection beyond the project time was financially supported by the Helmholtz Centre for Environmental Research (UFZ).

José Andrés López-Tarazón is in receipt of a Marie Curie Intra-European Fellowship (FLOODHAZARDS Project, PIEF-GA-2013-622468; 7th EU Framework Programme). José Andrés López-Tarazón and Ramon J. Batalla acknowledge the support from the Economy and Knowledge Department of the Catalan Government through the Consolidated Research Group 2014 SGR 645 (RIUS-Fluvial Dynamics Research Group).

Special thanks are due to the Ebro Water Authorities and its "Sistema Automático de Información Hidrológica y de Comunicación Fónica" (SAIH) for their permission to install the measuring equipment at the Capella gauging station, collaborative support during the investigation, providing field assistance and much useful data. In addition, the authors gratefully acknowledge the local communities and land owners for property access, cooperation and the permission to install observation systems.

For their efforts and commitment we want to thank the numerous students, interns and student assistants who were involved in field and lab work. Within the project, several Bachelor and Master's theses were realized, especially the work of Charlotte Wilczok contributed directly to the understanding and improvement of the presented dataset. We warmly thank all our colleagues from the Universities of Lleida and Potsdam, the German Research Centre for Geosciences (GFZ) and the Forest Technology Centre of Catalonia, including former project members and technicians, who helped in designing, installing and maintaining the monitoring systems and field stations.

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
