# Peer review of "Water and sediment fluxes in Mediterranean mountainous regions: Comprehensive dataset for hydro-sedimentological analyses and modelling in a mesoscale catchment (River Isábena, NE Spain)"

_Earth System Science Data, 2017_

## Referee Comment (RC1) · Anonymous Referee #1 · 7 Dec 2017

General comments: The paper by Till Francke et al. presents a dataset of hydro-sedimentary and meteorological data in a Mediterranean mesoscale catchment of North-East Spain (the Isábena catchment, 445 km2) comprising 5 instrumented sub-catchments ranging from 25 km2 to 146 km2. The dataset covers the period 2010-2016. It is interesting because there are few observation systems focused on discharge and suspended sediment fluxes in mountain mesoscale catchments in the Mediterranean region. The data provide mainly from the SESAM project and also from the Uni-

versity of Lleida and the SAIH operational network. This kind of data is very demanding to collect over a period of several years in such a network of hydrosedimentary stations since moutainous rivers are very dynamic geomorphological objects. Hydrometric and suspended sediment monitoring require regular observations in the field (gaugings, sampling) and significant amounts of samples to be collected after floods and analyzed in the lab. For the precipitation forcing, the authors have included data a network of 12 rain gauges deployed during the SESAM project and data from 2 rain gauges managed by the University of Lleida as well as 6 rain gauges managed by the SAIH operational network. For the other meteorological variables, the data come from the operational stations of the SAIH. All the data presented in the paper are easily accessible via the link proposed by the authors: http://doi.org/10.5880/fidgeo.2017.003. The data are in public access. However, the link to the CUASHI database does not work (http://hydroportal.cuahsi.org/isabena/cuahsi_1_1.asmx?WSDL). Overall, my opinion on this paper is contrasted. I think there is a potential, but the dataset selected seems to be of variable quality and some parts are missing, especially on the spatial descriptors. The authors highlight the value of this dataset for the evaluation of hydrosedimentary models, but no information is provided in the paper on the physical characteristics of soils to describe the hydrology of the catchment. The only information available are the spectral properties of surface soil samples used for the fingerprinting of suspended sediment. Similarly, there is no DTM or land use map provided while they are required to apply a distributed model. Regarding the presentation of the observation network, I did not find all the information I needed in terms of maps and tables to get a precise understanding of the instruments in place and their location. Figure 1C should be improved and completed. It is quite requiring for the reader to locate the points of measurements and what is really measured at each location. At the moment, it is necessary to enter deeply in the dataset to extract this information. In Figure 1C for example, there are too many red dots compared to the available dataset, so we get lost. I counted 18 red dots, while there are 6 rain gauges from the SAIH available in the accessible dataset. Furthermore I counted 11 black dots while 12 rain gauges are

listed in the paper for the SESAMII project. It is not easy to know where the different meteorological variables are measured. It would be important to find a way to add the names of the measurements points or at least a reference. It would also be nice to distinguish between "research" data and operational data in the Tables. Table 2 should be completed as well with the drainage area of the sub-catchments, the name of the rivers on which the measuring stations are located and the instrumentation deployed at each measuring station as it is the core of the dataset. It would be nice to add information on these stations in the dataset such as photos and the bathymetry of the cross-sections. I wonder the relevance of the choice of the selected period (2010-2016) presented in this study. I have the impression that this choice depends mainly on the deployment of the rain gauge network of the SESAMII project. However, the authors do not discuss about the density of rain gauges in relation to the size of the Isábena catchment and the gradient of altitude to determine if such an observation network is suitable to catch the spatial variability of rainfall in this mountainous Mediterranean environment. My feeling is that the density of the rain gauge network is variable over the catchment area and therefore the spatial variability of rainfall is only partially taken into account. Looking in more detail at the time series of precipitation, discharge and suspended sediment concentration, I realized that there was a decrease in data quality and completeness from the end of 2013: more gaps are visible in the time series of discharge and suspended sediment after this date and the sediment samples are not so well distributed in time to cover completely the floods. When reading the articles already published by the same authors about the Isábena catchment, I realized that the period 2005-2010 seemed of better quality in terms of discharge and suspended sediment times series compared to the 2014-2016 period. So, why not include in this paper the period 2005-2013 and think about the relevancy of maintaining the period 2013-2016? Continuous time series are particularly required to establish water and sediment budget. That would be fine also to add some information on the diversity of events observed during the period in terms of return period (at least for SAIH precipitation rain gauges and discharge at the Capella station where there are longer time series).

More specific comments on the dataset: Precipitation data: I would propose to organize the data in sub-directories according to the producers (SAIH, U. de Lleida, SESAMII). There is too much heterogeneity in the format of the data. It is certainly interesting to provide data at the time step of the tipping bucket but it does not seem enough to me. The data should also be provided at a fixed time step, common to all the measurement points (at least the research rain gauges): 1 'or 5'. This would highlight possible periods of gaps and make easier for people who are not used to manage files at the time step of the tipping bucket to have a quick overview of the data. Times series should also all start and end on common dates (example: 01/01/2010 00:00 and 31/12/2016 23:55) and include all the time steps (the lacking values should be indicated by -9999). For Villacarli, there seems to be 2 rain gauges at the same place. It is not explained clearly in the text and it is probably the reason why there are 11 black dots in Figure 1 instead of 12.

Other meteorological data: It is not clear where these variables were measured by reading the paper and looking Figure 1C. Is their location relevant for the Isábena catchment which is located in a region of important gradient of altitudes? There seems to be mainly temperature measuring stations compared to the other variables but are they able to account for the effect of altitude ?

Calculated discharge data: There are 6 stations including 5 stations belonging to the SESAMII project and 1 station ruled by SAIH (Capella). Temporal resolutions are 1, 5 or 15 minutes. Regarding the data at the time step of 5 min, I was surprised to find out that the data are not necessarily stored for multiples of 5 min (3, 8, 13, 18, 23. . . instead of 0, 5, 10, 15, 20, 25, 30, 35. . .) and it can change over time. In addition, there are sometimes changes in time steps. It would be appropriate to build files with fixed time steps that also include gaps with value of -9999. For Villacarli, there is a change from a time step of 5 min to 1 min on 22/11/2011 within the same time series. As for rainfall, time series should all start and end on common dates. Regarding the stage-discharge rating curves, I questioned the difficulty of maintaining the ratings in

gravel bed rivers which are considered as moving-bed rivers. The authors make no comment on this difficulty while it constitutes an obstacle for the scientific community at the moment. How are managed the problems of successive shifts in rating curves resulting from the changes in cross section geometry? The BaRatin tool (Le Coz et al., 2014) that was used in this study does not handle shifts in rating curves. It should be applied to independent periods associated with a stable rating curve. In addition, I wanted to carry out a few verifications of discharge contribution from sub-catchments to the whole Isabena catchment at the scale of the flood event. I selected randomly two events: the 03/04/2014 and the 03/11/2015 but there were too many gaps and it was not possible to perform such a test.

Discharge measurements: These data must be clearly separated from the calculated discharge time series since they correspond to direct measurements. The discharge measurements should not appear in Table 1. In addition, in the "discharge_metering.csv" file, it is essential to add the value of water level for each gauging since it is part of the measurement. The authors should be careful since there are some negative values in this file? Additionnaly, since there are no gaugings at very high water, it is important to explain according to which hypothesis the rating curves were extrapolated? If it is derived from BaRatin, the different hydraulic controls should be listed in a table and the cross-section should be added in the dataset.

SSC data: generally there are less samples after mid-2013 and lower concentration values, what is the reason for such a behaviour? In the methodology section of the paper, it is not explained how turbidity-SSC rating curves were derived. Are they general rating curves or specific to each flood event? How are the data processed in the absence of collected samples? Capella_turbidity_1.csv: bad signal from June 2014 Capella_turbidity_2.csv: negative signal permanently

Specific remarks on the text: Some bibliographic references are not present in the list while they are quoted in the text. p.6 l.9-11: not clear the number of weather stations. p.7 l.14: on what criteria are the data criticized? p.7 l.15-19: is there a possible expla-

nation for this phenomenon? p.9 l.22: replace Vwater with Vmixture p.12 l.19 : replace http://doi.org/10.5880/fidgeo.2017.03 with http://doi.org/10.5880/fidgeo.2017.003

---

## Short Comment (SC1) · 12 Dec 2017

Many thanks to Referee #1 for the thorough assessment and the numerous suggestions. Before replying point-by-point, we'd like to quickly clarify an issue that could potentially also help the second reviewer: Ref#1 claimed that the supplied URL (http://hydroportal.cuahsi.org/isabena/cuahsi_1_1.asmx?WSDL) would not work. This is probably a misconception, as it does work, but is not a HTTP-Service to be

viewed with a browser. Instead, this is the URL required for the CUAHSI-API services. This API allows tailored data access with numerous other tools (e.g. HydroDesktop, WaterML R-package). These tools offer convenient interactive ways for "understanding of the instruments in place and their location" as requested by the Ref#1. When preferring not to install any software, we recommend HydroClient accessed via http://data.cuahsi.org/. Clicking "Data Services" and searching for "Isábena" provides various options to explore our database. We hope that this way of access does already answer many of the raised questions.

―――――――――――――――――――――

---

## Referee Comment (RC2) · Anonymous Referee #2 · 18 Jan 2018

The manuscript presents a dataset on water and sediment fluxes in a Mediterranean mountainous region. The dataset is large and well organise (specific comments hereafter). I believe that the dataset is valuable for the scientific community and I recommend the manuscript for publications. Nonetheless, the authors have used parts of the dataset in (at least) eight other publications. While this might not be a concern, I do think that there is room for improving the manuscript (especially the one figure and the two tables) to provide a better overview of the available information (suggestions

hereafter).

I was able to download the data without problems from GFZ data services. It is also possible to easily find and download the datasets in cuahsi. Moreover, as claimed by the authors, the data is stored following the cuahsi rationale.

GENERAL COMMENTS

GIS data is often needed in many studies. I was wondering if it would be possible to add a section/table providing information on the data that is available (e.g. maps, resolution, source, links, etc) (see Table 3 in Nord et al., 2016). Also including details and link to the data published by Foerster et al 2015. I know that in many cases gis data might not be freely available. Nonetheless, I would certainly appreciate to know what is available and where.

After reading the last paragraph of the introduction, describing publications that have used parts of the database, I have noticed that some publications are missing (e.g. López-Zarazón et al 2009). I think it is important to list all the studies and briefly explain the objectives/results of these studies and the data that they have each used. It would be useful to summarised the information in a table.

Please, improve Figure 1. Where are located the 'main badland areas'? They are difficult to see despite its major importance as sediment sources. I would encourage the authors to find a better display. I would slightly change the symbols indicating the meteo and gauges stations, they can not be distinguished when not printing in color. Have the authors consider adding some pictures of the catchment? I think it could be useful. Also, it is not possible to quickly see in the map all the stations and what it is measured in each (i.e. a clear link between figure 1 and table 1 does not exist at the moment). A land use map and geology/pedology map would be helpful (and even essential if the data is to be used for distributed modelling). Also display the location of the soil sampling points (I would add another figure close to section 3.4) to give the reader a first idea on distribution within the catchment.

I would encourage the authors to improve Table 1, too. I think it is important to have a complete overview of the data made available. In its current way, the reader needs to open the data source or read the complete manuscript to know where is every variable measured, during which period, with which instrument (include instrument type, model, etc). Which instruments are installed in the official stations? I would use the word 'sporadic' or 'punctual' instead of 'intermittent', and 'reflectance spectra' instead of 'spectra'. In the database you use '_wavelength_nm_X' instead of 'wavelength_nm_X". I guess you want these variables to appear at the end of the list in the 'variables table'. Please, be consistent.

The variable 'ssc_sampled' includes samples collected manually, and samples collected with an automatic sampler. To my understanding, this is a mistake because automatic samplers are subject to the uncertainties associated with the sampling apparatus. A calibration between cross-sectional manual samples and automatic samplers for each site should be provided.

Section 3.1 Explain how the rain gauges have been calibrated and controlled during the measuring period. You mention in the text that 'Snowmelt' occurs (Page 4-line 10). Is snow quantified? Are the rain gauges heated? Do you think that trends observed in section 3.1.2 are real? Or are they due to instrument malfunctioning? I have plotted a couple of time series and I do not see decreasing trends. Have you perform statistical tests? Please, provide more evidence. The authors mention that there are large pluviometric gradients. I guess it is possible to capture such gradients as many measuring points are available, but please, provide some evidence. I would appreciate to visualize the discharge rating curves in the paper. Would it be possible to add a figure with the six rating curves, associated uncertainties and maximum measured water stage values (to have an idea of the extrapolation range). Also, distinguishing the different methods used to measure discharge, i.e. velocity-area, dilution methods. . . the method used to determine the associated uncertainties should be detailed. We are refered to Lopez-Tarazon et al 2010 to have more information on the rating curve, despite that the data

presented was collected from 2010 to 2016.

Provide evidence that 'average SSC in the section can generally be assumed to be acceptable due to the observed high flow turbulence'. As I mentioned before, samples collected manually and with automatic samplers should be presented separated, and the authors should prepare a figure comparing ssc from manual/automatic samples (for each sampling site). I would also suggest to add a figure with the rating curves to estimate ssc from turbidity (including uncertainty ranges and distinguishing how the water samples where collected: manually/automatic sampler).

Have other parameters (together with spectral reflectance) hae been measured in the soil/sediment samples? If the answer is yes, please explain that in the text even if the authors have chosen not to make the data available. At what height were mounted the light source and the senor for measuring the spectra in the field? The authors mention that spectras measured in loose material should not be compared to those measured in filters (please add references in page 12-line 6-10). I wonder if the authors have consider transferring the loose materials into filters, to measure the spectra and be able to compare the soil and sediment information. I would also appreciate a plot showing the measuring gaps for each data set (i.e. variable/station). I would encourage the authors to find a visual way to show the quality of the collected data in a plot.

In the variables table. Are water stage and discharge data 'average' data or punctual? Also, why variable name for reflectance is 'albedo'? elevation data could added for the soil sites. Almost no information is provided for the soil sampling sites in the database, only land use type (e.g. 'grassland'), could addition information be added (e.g. soil type, organic matter content,...)

SPECIFIC COMMENTS - You refer to the study site as 'dryland region' (e.g. Page 2-Line 5, Page 2-line23). Having in mind that the Isábena catchment has a Mediterranean climate, I wonder if this is correct to use this terminology. If I remember it well, other authors use terms as 'humid Mediterranean catchments with badland'. -

Page 2, lines 10-11. I disagree on the fact that there has been little research ('or 'much less research') on relevant landscape components for water and sediment management. Please, could you explain this better or add references. - Page 2, line 18. Aren't Nord et al. 2016 presenting sediment data (i.e. hydro-meteo-sedimentological data?)? Please, check. You then say that in the following line that you present an hydro-sedimentological dataset, what about the meteo? Please, be consistent. - Page 2, lines 27-28: Just a suggestion. I would like to read here the measurement period and the catchment area. - Page 4-lines 8-9. Which data did you use to calculate these average values? Which period? - Page 4-line 10. Flow regime IS characterized or flow regimes are characterised. - Page 4. Sometimes you use the term 'mean' and others 'average'. I would suggest to be consistent. - Page 4-line 12. I am confused with the term 'mean annual discharge'.... Do you mean 'mean discharge' or 'mean punctual discharge' or 'mean instantaneous discharge'? which data period have you use to estimate these values? - Page 4, last paragraph. Please, revise punctuation. - Page 5-line 21-22. Why data from 2004 to 2010 is not included in the dataset? - Page 6, lines 3-4. Please, reformulate the sentence.

---

## Author Comment (AC1) · 14 Feb 2018

On behalf of all co-authors, we thank the two referees for their thorough assessment of the manuscript. Their detailed comments will help us to correct the remaining inconsistencies and improve the manuscript.

*To the editor:*
Both reviewers requested numerous additional information and figures, namely:
- Spatial catchment descriptors / maps[x]
- Photos of cross sections*
- Photos of catchment(*)
- Table on return periods*
- Plots of coverage of time series*
- Table of third-party data sources*
- Table of publications using the data*
- Map of soil sample locations[x]
- Discharge rating curves(*)
- Turbidimeter rating curves(*)
- Plot with data quality of time series[x]

These amendments would increase the page count of the manuscript notably (~ 8 pages). Given the mission statement of ESSD[1], we are unsure if this is appropriate. We kindly ask for guidance in that regard from the editor.
Our recommendations above:
*: could be added;
(*): possible, but probably not necessary
[x]: impossible or unnecessary
(see detailed comments below)

[1] " […] the publication of articles on original research data" [with] "planning, instrumentation, and execution of experiments or collection of data. Any interpretation of data is outside the scope of regular articles." (https://www.earth-system-science-data.net/about/aims_and_scope.html).

As requested, we to reply point-by-point to the reviewer comments:

Anonymous Referee #1

General comments: The paper by Till Francke et al. presents a dataset of hydro-sedimentary and meteorological data in a Mediterranean mesoscale catchment of North-East Spain (the Isábena catchment, 445 km2) comprising 5 instrumented sub-catchments ranging from 25 km2 to 146 km2. The dataset covers the period 2010-2016. It is interesting because there are few observation systems focused on discharge and suspended sediment fluxes in mountain mesoscale catchments in the Mediter-ranean region. The data provide mainly from the SESAM project and also from the University of Lleida and the SAIH operational network. This kind of data is very demanding to collect over a period of several years in such a network of hydrosedimentary stations since moutainous rivers are very dynamic geomorphological objects. Hydrometric and suspended sediment monitoring require regular observations in the field (gaugings, sampling) and significant amounts of samples to be collected after floods and ana-lyzed in the lab. For the precipitation forcing, the authors have included data a network of 12 rain gauges deployed during the SESAM project and data from 2 rain gauges managed by the University of Lleida as well as 6 rain gauges managed by the SAIH operational network. For the other meteorological variables, the data come from the operational stations of the SAIH.

AC1: We thank Referee #1 for the positive assessment.

All the data presented in the paper are easily accessible via the link proposed by the authors: http://doi.org/10.5880/fidgeo.2017.003. The
data are in public access. However, the link to the CUASHI database does not work (http://hydroportal.cuahsi.org/isabena/cuahsi_1_1.asmx?WSDL).
AC2: As already mentioned in the short comment "**SC1**: 'Quick reply and clarification of access',
there has apparently been a misunderstanding in the mode of data access. As testified by Referee #2, the data access does work. However, we will clarify this issue with some more explanation in the revised version of the manuscript and provide an additional URL for the landing page (http://hiscentral.cuahsi.org/pub_network.aspx?n=5622)  as opposed to the URL of the API in section 5.

Overall, my opinion
on this paper is contrasted. I think there is a potential, but the dataset selected seems to be of variable quality and some parts are missing, especially on the spatial descriptors. The authors highlight the value of this dataset for the evaluation of hydrosedimentary models, but no information is provided in the paper on the physical characteristics of soils to describe the hydrology of the catchment. The only information available are the spectral properties of surface soil samples used for the fingerprinting of suspended sediment. Similarly, there is no DTM or land use map provided while they are required to apply a distributed model.
AC3: We acknowledge that the data set is not free of gaps. Considering that instruments and protocols have been maintained constant, we are not sure what exactly Ref#1 means with "variable quality".
We agree that the data we published is not a complete package that allows starting modeling studies right away. Depending on the employed model and purpose, various other data may be necessary (topography, soils, vegetation, …). We do not claim to provide these, nor do we have the legal right to distribute these.
Instead, we limit the publication to the data that were acquired solely by *our own* activities or (in the case of the SAIH data) complement our measurements and cannot otherwise be acquired freely.
From the technical point of view, the CUAHSI structure is unsuitable for storing geospatial data. Thus, the employed primary storage site (http://hydroportal.cuahsi.org/ ) cannot be used to hold the entire dataset.
We would therefore prefer to keep this original scope of the work. However, we will assemble a table listing third-party data sources, even if we cannot guarantee the persistence of these sources.

Regarding the presentation of the observation network, I did not find all the information I needed in terms of maps and tables to get a precise understanding of the instruments in place and their location. Figure 1C should be improved and completed. It is quite requiring for the reader to locate the points of measurements and what is really measured at each location. At the moment, it is necessary to enter deeply in the dataset to extract this information. In Figure 1C for example, there are too many red dots compared to the available dataset, so we get lost. I counted 18 red dots, while there are 6 rain gauges from the SAIH available in the accessible dataset. Furthermore I counted 11 black dots while 12 rain gauges are listed in the paper for the SESAMII project. It is not easy to know where the different meteorological variables are measured. It would be important to find a way to add the names of the measurements points or at least a reference.
AC4: As mentioned in AC2, this information is readily accessible via the web interface much better than any static map could ever provide. We'd like to encourage Ref#1 to take advantage of these capabilities. Nevertheless, Ref#1 is right about some inconsistencies of Fig. 1C, which will be improved (see also AC 32).

It would also be nice to distinguish between "research" data and operational data in the Tables.
AC5: We will add this information to Table 1.
According to the CUAHSI scheme, the different source of the data can also be abstracted from the documented meta-data (Table "Sources").

Table 2 should be completed as well with the drainage area of the sub-catchments, the name of the rivers on which the measuring stations are located and the instrumentation deployed at each measuring station as it is the core of the dataset.
AC6: We will add drainage area to Table 2.
The names of the stations are in accordance to the scheme used by the first respective research in this area (Verdú 2003) and the majority of the subsequent publications. The names of the respective creeks differ, depending on which maps are consulted and which language is used (Castellano, Catalan, Aragonese) and are therefore of little use here. We will complement the name of the instruments. According to the CUAHSI scheme, this information is also contained in the meta-data (Table "Sources").

Verdú, J. M. (2003), Análisis y modelización de la respuesta hidrológica y fluvial de una extensa cuenca de montaña mediterránea (río Isábena, Pre-Pirineo), Universitat de Lleida, Lleida, Catalonia, Spain, Available from: http://www.tdx.cat/TDX-0630107-193135.

It would be nice to add information on these stations in the dataset such as photos and the bathymetry of the cross-sections.
AC7: For reasons of conciseness, we had refrained from adding photos. Following the suggestions of both reviewers, we will add them.
We are likewise ready to publish cross-section information. However, as mentioned in AC3, this geospatial data cannot be accommodated in the primary storage site (*http://hydroportal.cuahsi.org/isabena/cuahsi_1_1.asmx?WSDL* ) and could only be placed onto the alternative mirror (http://doi.org/10.5880/fidgeo.2017.003 ). We are somewhat concerned that this compromises the correspondence of the two mirrors of the same dataset, though.

I wonder the relevance of the choice of the selected period (2010-2016) presented in this study. I have the impression that this choice depends mainly on the deployment of the rain gauge network of the SESAMII project.
AC8: Ref#1 is right in assuming that the selected period is governed by the availability of the denser rain gauge network starting from the beginning of the SESAM-II-project. While some data (Q, P) have also been collected before this phase, the respective methods (instruments, resolution, locations, protocols) differed. We therefore chose to present the consistent part of the dataset only.

However, the authors do not discuss about the density of rain gauges in relation to the size of the Isábena catchment and the gradient of altitude to determine if such an observation network is suitable to catch the spatial variability of rainfall in this mountainous Mediterranean environment. My feeling is that the density of the rain gauge network is variable over the catchment area and therefore the spatial variability of rainfall is only partially taken into account.
AC9: We fully share these concerns. The raingauge network represented the best effort to supplement the official network and capture rainfall heterogeneity given the constraints of instrument budget, manpower for maintenance, accessibility and instrument safety. Still, field observations suggest that especially convective events may poorly be recorded by the raingauges. We intended to quantify this using rainfall radar data. Unfortunately, public access to AEMET rainfall data was discontinued three months after we became aware of it. Moreover, some preliminary tests suggest that the distance to the radar stations (> 100 km for Barcelona and Zaragoza) and pronounced topography had not warranted highly reliable data.

Looking in more detail at the time series of precipitation, discharge and suspended sediment concentration, I realized that there was a decrease in data quality and completeness from the end of 2013: more gaps are visible in the time series of discharge and suspended sediment after this date and the sediment samples are not so well distributed in time to cover completely the floods. When reading the articles already published by the same authors about the Isábena catchment, I realized that the period 2005-2010 seemed of better quality in terms of discharge and suspended sediment times series compared to the 2014-2016 period. So, why not include in this paper the period 2005-2013 and think about the relevancy of maintaining the period 2013-2016? Continuous time series are particularly required to establish water and sediment budget.

AC10: Ref#1 correctly noted that data density and gaps in the time series differ in time. This is mainly a result of instrument failures, violent events and vandalism versus manpower for maintenance. We do not agree that this affect data *quality* per se.

As stated in AC8, data were already recorded prior to 2010 during a PhD-phase. While the respective manpower provided higher density of SSC-samples at some stations, the methodological details differed. We therefore chose to present the consistent part of the dataset only.

That would be fine also to add some information on the diversity of events observed during the period in terms of return period (at least for SAIH precipitation rain gauges and discharge at the Capella station where there are longer time series).

AC11: We will try to obtain the required long-term data from SAIH and provide this information.

More specific comments on the dataset: Precipitation data: I would propose to organize the data in sub-directories according to the producers (SAIH, U. de Lleida, SESAMII).

AC12: The (precipitation) data can be arranged in any desired manner using the functionality of the CUAHSI database. Anyway, we have arranged the data files in the ZIP-archives in the requested manner.

There is too much heterogeneity in the format of the data. It is certainly interesting to provide data at the time step of the tipping bucket but it does not seem enough to me. The data should also be provided at a fixed time step, common to all the measurement points (at least the research rain gauges): 1 'or 5'. This would highlight possible periods of gaps and make easier for people who are not used to manage files at the time step of the tipping bucket to have a quick overview of the data.

AC13: The data format strictly adheres to CUAHSI specifications, no other formats are used. However, temporal resolution of the time series differ (see Table 1).

Tipping bucket data (i.e. breakpoint data) provides the maximum temporal resolution inherent in the data. We chose to preserve this to allow the analysis of short-term rainfall intensities. These data can be aggregated to any desired resolution, but we did not include this redundant data for the sake of conciseness. A respective R-script can be found in the digital annex.

Times series should also all start and end on common dates (example: 01/01/2010 00:00 and 31/12/2016 23:55) and include all the time steps (the lacking values should be indicated by -9999).

AC14: We prefer to include the full amount of data available until publication.

For regular time series, missing values ("not recorded") are simply absent; while only invalid data but still potentially of interest ("recorded but corrupted") are masked with -9999 to conserve storage space.

For breakpoint data, periods of missing data are denoted by -9999 at start and end, which can be dealt with by the script mentioned in AC13.

For Villacarli, there seems to be 2 rain gauges at the same place. It is not explained clearly in the text and it is probably the reason why there are 11 black dots in Figure 1 instead of 12.
AC15: The subcatchment Villacarli hosts four raingauges: Via1, Via2, Villacarli_Bridge_1 and Villacarli_Bridge_2 (see table Sites). The latter two are replicates (only few meters apart), and therefore not distinguishable in Fig. 1. All site data is fully available in the CUAHSI Table Sites.

Other meteorological data: It is not clear where these variables were measured by reading the paper and looking Figure 1C. Is their location relevant for the Isábena catchment which is located in a region of important gradient of altitudes? There seems to be mainly temperature measuring stations compared to the other variables but are they able to account for the effect of altitude ?
AC16: For temperature, radiation and humidity, the included SAIH stations (see Fig. 1) constitute the best available data source. With elevations ranging from 450 to 1900 m asl (see table Sites), they cover a large part of the elevation range. Depending on the intended application, the discernible altitudinal gradients may be used to infer values at unmonitored locations.

Calculated discharge data: There are 6 stations including 5 stations belonging to the SESAMII project and 1 station ruled by SAIH (Capella). Temporal resolutions are 1, 5 or 15 minutes. Regarding the data at the time step of 5 min, I was surprised to find out that the data are not necessarily stored for multiples of 5 min (3, 8, 13, 18, 23, . . . instead of 0, 5, 10, 15, 20, 25, 30, 35 . . .) and it can change over time. In addition, there are sometimes changes in time steps. It would be appropriate to build files with fixed time steps that also include gaps with value of -9999. For Villacarli, there is a change from a time step of 5 min to 1 min on 22/11/2011 within the same time series.
AC17: We preferred to retain the maximum resolution recorded, as temporal aggregation can always be done, while disaggregation cannot (see also AC 13, AC14).

As for rainfall, time series should all start and end on common dates.
AC18: See AC14.

Regarding the stage-discharge rating curves, I questioned the difficulty of maintaining the ratings in gravel bed rivers which are considered as moving-bed rivers. The authors make no comment on this difficulty while it constitutes an obstacle for the scientific community at the moment. How are managed the problems of successive shifts in rating curves resulting from the changes in cross section geometry? The BaRatin tool (Le Coz et al., 2014) that was used in this study does not handle shifts in rating curves. It should be applied to independent periods associated with a stable rating curve.
AC19: We agree with Ref#1 that obtaining the waterstage-discharge relationships constitutes a significant uncertainty in the discharge data. The respective approach has been outlined in section 3.2.1 already, but will be complemented like this
"For gauges Carrasquero, Ceguera and Villacarli (see Figure 1), changes in the geometry of the streambed were observed after heavy floods. The levels of zero-flow and time of its changes have been reconstructed manually and interpolated linearly. The respective difference to the effective flow level has been added as offset information in the database. Thus, both the absolute water level $h_{abs}$ and the effective water level $h_{eff}$ can be retrieved with

$$h_{eff} = h_{abs} - \text{offset}$$

For the subsequent steps of converting water stage into discharge, $h_{eff}$ was used."
To acknowledge the additional uncertainty, following sentence will be added to section 3.2.3:

"For the gauges with correction of zero-flow level (see section 3.2.1), this implies the assumption that the remaining hydraulic controls (lateral restrictions by bridge walls, bed roughness and slope) remained unaltered, which seems a reasonable assumption according to observations. Still, the resulting uncertainty in discharge contained in the database will probably be optimistic."

In addition, I wanted to carry out a few verifications of discharge contribution from sub-catchments to the whole Isabena catchment at the scale of the flood event. I selected randomly two events: the 03/04/2014 and the 03/11/2015 but there were too many gaps and it was not possible to perform such a test.

AC20: Without doubt, Ref#1 is correct that there will be several events when at least one gauge failed to record data. This is indeed regrettable, but cannot be changed.
An illustration of the coverage of time series will be added (see AC49).

Discharge measurements: These data must be clearly separated from the calcu-lated discharge time series since they correspond to direct measurements. The discharge measurements should not appear in Table 1.

AC21: We do not share this concern: Both variables ("discharge_*" and "discharge_metering") are clearly distinguished from another (see Table 1 and Table "Variables").
Generally, all contained variables cover a spectrum from "more direct" (e.g. temperature) to "less direct" (discharge from water stage from capacitive voltage) measurements. We do not see a clear and meaningful way of distinction here.

In addition, in the "discharge_metering.csv" file, it is essential to add the value of water level for each gauging since it is part of the measurement. The authors should be careful since there are some negative values in this file?

AC22: Corresponding water levels can be found in the variable "water_stage_reading" (see table water_stage_reading.csv). We agree that these two files belong together conceptually. However, technically they cannot be merged without violating the CUAHSI format. Negative values can have resulted when water stage was very close to zero flow level. They are not problematic, as they constitute a relative reference only.

Additionnaly, since there are no gaugings at very high water, it is important to explain according to which hypothesis the rating curves were extrapolated? If it is derived from BaRatin, the different hydraulic controls should be listed in a table and the cross-section should be added in the dataset.

AC23: We made great effort to cover also high flow conditions with measurements. Of course, the very highest water stages could not be accompanied by discharge meterings. We expect no change of hydraulic control for the weir-shaped cross sections (Capella, Cabecera), the confinements of the bridge walls (Ceguera, Carrasquero, Villacarli) nor the pipe of corrugated-metal (Lascuarre), allowing the extrapolation of the rating curve. A respective explanation will be added to section 3.2.3.
As commented in AC7, we would rather refrain from adding Baratin-files to the data base, but will do so in the digital annex, if desired (see also AC41).

SSC data: generally there are less samples after mid-2013 and lower concentration values, what is the reason for such a behaviour?

AC24: Emptying the automatic samplers required personal attendance, which was not equally available during the entire timespan. Since samplers were configured also to sample lesser floods, this may have resulted in missing subsequent events (potentially, with higher concentrations). Thus, the lower sampling frequency may create the impression of lower SSCs. Similarly, this may also be a pure coincidence in the face of the highly variable dynamics of the catchment.

In the methodology section of the
paper, it is not explained how turbidity-SSC rating curves were derived. Are they general rating curves or specific to each flood event? How are the data processed in the
absence of collected samples?
AC25: Section 3.3.2 holds the requested information, namely
"Using the abovementioned lab-samples, the valid turbidimeter data was converted to *SSC* using
quadratic regression, constituting separate time series in the database." We will add
"These rating curves are considered invariant for the time period."
Following the concept of rating curves, the "absence of collected samples" is the rule rather than the
exception for the entire time series. As the turbidimeter rating curves are assumed invariant (offset
nonwithstanding, see 3.3.2), they can also be used during periods of no samples, because sufficient
samples have been collected during other timespans.

Capella_turbidity_1.csv: bad signal from June 2014
Capella_turbidity_2.csv: negative signal permanently
AC26: We appreciate the thorough checking. Indeed, this file was flawed, as it did not contain the
drift-corrected signal (see section 3.3.2). These files will be replaced.

Specific remarks on the text: Some bibliographic references are not present in the list
while they are quoted in the text.
AC27: The missing references will be added.

 p.6 l.9-11: not clear the number of weather stations.
AC28: This refers to Table 1, where the number of stations / sites is listed in column 2. We do not
understand this comment, please explain.

p.7 l.14: on what criteria are the data criticized? p.7 l.15-19: is there a possible explanation
for this phenomenon?
AC28: The passage describes the data obtained from SAIH stations. We chose not to alter these data.
Still, there are some apparent peculiarities, which could not be clarified despite checking back with
the SAIH people. We think that these qualitative remarks can be of help to the potential user of the
data to decide if this poses an issue or not for them (also see AC40).

p.9 l.22: replace Vwater with Vmixture
AC29: The specific equation correctly describes how SSC was calculated. We are aware that –
depending on the intended use – Vmixture could also be used as the denominator. To clarify the
convention we used, this equation was explicitly added.

p.12 l.19 : replace
http://doi.org/10.5880/fidgeo.2017.03 with http://doi.org/10.5880/fidgeo.2017.003
AC29: Done.

Anonymous Referee #2

The manuscript presents a dataset on water and sediment fluxes in a Mediterranean
mountainous region. The dataset is large and well organise (specific comments here-
after). I believe that the dataset is valuable for the scientific community and I recommend the manuscript for publications. Nonetheless, the authors have used parts of the dataset in (at least) eight other publications. While this might not be a concern, I do think that there is room for improving the manuscript (especially the one figure and the two tables) to provide a better overview of the available information (suggestions I was able to download the data without problems from GFZ data services. It is also possible to easily find and download the datasets in cuahsi. Moreover, as claimed by the authors, the data is stored following the cuahsi rationale.

GENERAL COMMENTS

GIS data is often needed in many studies. I was wondering if it would be possible to add a section/table providing information on the data that is available (e.g. maps, resolution, source, links, etc) (see Table 3 in Nord et al., 2016). Also including details and link to the data published by Foerster et al 2015. I know that in many cases gis data might not be freely available. Nonetheless, I would certainly appreciate to know what is available and where.

AC30: (see AC3): we will assemble a table listing third-party data sources, even if we cannot guarantee the persistence of these sources.

After reading the last paragraph of the introduction, describing publications that have used parts of the database, I have noticed that some publications are missing (e.g. López-Zarazón et al 2009). I think it is important to list all the studies and briefly explain the objectives/results of these studies and the data that they have each used. It would be useful to summarised the information in a table.

AC31: "López-Tarazón et al 2009" is only cited in section 2, as it precedes the timespan of the data collection associated to the manuscript. However, we will add the proposed table listing the references that used (parts of) the data.

Please, improve Figure 1. Where are located the 'main badland areas'? They are difficult to see despite its major importance as sediment sources. I would encourage the authors to find a better display. I would slightly change the symbols indicating the meteo and gauges stations, they can not be distinguished when not printing in color.

AC32: The figure will be improved accordingly (see also AC4).

Have the authors consider adding some pictures of the catchment? I think it could be useful. Also, it is not possible to quickly see in the map all the stations and what it is measured in each (i.e. a clear link between figure 1 and table 1 does not exist at the moment).

AC33: Agreed, see AC32 and AC7.

 A land use map and geology/pedology map would be helpful (and even essential if the data is to be used for distributed modelling).

AC34: Please see AC3.

Also display the location
of the soil sampling points (I would add another figure close to section 3.4) to give the reader a first idea on distribution within the catchment.

AC35: We will add a reference to Brosinky et al, 2014a, where the requested figure is available.

I would encourage the authors to improve Table 1, too. I think it is important to have a complete overview of the data made available. In its current way, the reader needs to open the data source or read the complete manuscript to know where is every variable measured, during which period, with which instrument (include instrument type, model, etc). Which instruments are installed in the official stations? I would use the word 'sporadic' or 'punctual' instead of 'intermittent', and 'reflectance spectra' instead of 'spectra'. In the database you use '_wavelength_nm_X' instead of 'wavelength_nm_X". I guess

you want these variables to appear at the end of the list in the 'variables table'. Please, be consistent.

AC36: We prefer Table 1 to remain a *summary* of the data, not an exhaustive list. In terms of the latter, we will add a reference to the respective CUAHSI tables that contains the full details, i.e. multiple variables with multiple instruments, etc., that would otherwise clutter Table 1 unnecessarily.

The variable 'ssc_sampled' includes samples collected manually, and samples collected with an automatic sampler. To my understanding, this is a mistake because automatic samplers are subject to the uncertainties associated with the sampling apparatus. A calibration between cross-sectional manual samples and automatic samplers for each site should be provided.

AC37: Indeed, "ssc_sampled" contains both manual and automatically collected samples. Both were collected at the side of the stream. We assume their representativity "to be acceptable due to the observed high flow turbulence and the predominantly fine sediment fractions" (see section 3.3.1). This was found with some test samples at the widest cross section. Therefore, we did not any calibration between automatic and manual samples. A respective explanation will be added to section 3.3.1.

Section 3.1 Explain how the rain gauges have been calibrated and controlled during the measuring period.

AC38: We will add "The raingauges have been calibrated in-situ with dripping bottles repeatedly after their set-up. As these calibrations did not differ notably, an invariant calibration was used for the entire time period."

You mention in the text that 'Snowmelt' occurs (Page 4-line 10).
Is snow quantified? Are the rain gauges heated?

AC39: The Davis rain gauges are not heated, snow cannot be quantified separately. However, recent analysis by Rottler (2017) indicated that snow accumulation in the tipping buckets is generally negligible. A respective comment will be added.
Rottler, E. (2017) "Implementation of a snow routine into the hydrological model WASA-SED and its validation in a mountainous catchment", unpublished MSc

Do you think that trends observed in section 3.1.2 are real? Or are they due to instrument malfunctioning? I have plotted a couple of time series and I do not see decreasing trends. Have you perform statistical tests? Please, provide more evidence.

AC40: "Global radiation time series show a slight but steady decrease in recorded values over time." (section 3.1.2, see also figure below) We tend to belief that these are sensor artefacts. We could not get any further explanations from our contact person. Still, we are somewhat reluctant to point a finger at the people there without further knowledge and prefer to stick with this neutral observation. Without excluding instrument artefacts, we see no point in a further statistical analysis (see also AC28)

[Figure]

The authors mention that there are large pluviometric gradients. I guess it is possible to capture such gradients as many measuring points are available, but please, provide some evidence.
AC40: Rainfall characteristics have been described by Verdú et al., 2006. The respective reference will be added.

I would appreciate to visualize the discharge rating curves in the paper. Would it be possible to add a figure with the six rating curves, associated uncertainties and maximum measured water stage values (to have an idea of the extrapolation range).
AC41: The requested figures can be reproduced with the provided input files for BaratinAGE (see also AC23). They are given below for convenience, for explanations, please consult the explanations in BaratinAGE.
We would rather prefer not to include them in the paper to keep the manuscript balanced and concise. Without linking them to the recorded water stages we see only limited added value in them.

[Figure]

Also, distinguishing the different methods
used to measure discharge, i.e. velocity-area, dilution methods the method used to
determine the associated uncertainties should be detailed.
AC42: All discharge meterings are in detail recorded with their respective method in Table
Methods and the supplemental material (hydro\quncertainties.xlsx, BarainAGE-files). A
respective reference will be added to section 3.2.2.

We are refered to Lopez-Tarazon et al 2010 to have more information on the rating curve,
despite that the data presented was collected from 2010 to 2016.
AC43: The gauge Capella consists of a stable broad-crested weir (see Table 2). Its rating
curve had been derived before 2010 and can be assumed invariant.

Provide evidence that 'average SSC in the section can generally be assumed to be
acceptable due to the observed high flow turbulence'. As I mentioned before, samples
collected manually and with automatic samplers should be presented separated, and
the authors should prepare a figure comparing ssc from manual/automatic samples
(for each sampling site).
AC44: We tested the representativity only with some tests in the beginning of the period (see
AC37). After that, no concomitant manual and automatic sampling was performed that would
allow such a comparison. We will try to reconstruct which of the samples were collected in
which manner, if possible.

I would also suggest to add a figure with the rating curves
to estimate ssc from turbidity (including uncertainty ranges and distinguishing how the
water samples where collected: manually/automatic sampler).

[Figure]

AC45: The requested figures can be reproduced with the files in the supplementary materials.

We would rather prefer not to include them in the paper to keep the manuscript balanced and concise.

We cannot provide uncertainty ranges, as the SSC-turbidity relationship displays pronounced heteroscedasticity, which violates the assumptions that are needed to calculate confidence bounds in linear regression.

Have other parameters (together with spectral reflectance) hae been measured in the soil/sediment samples? If the answer is yes, please explain that in the text even if the authors have chosen not to make the data available.

AC46: So far, no other properties of the collected samples have been analyzed. We envision further work on this issue.

At what height were mounted the
light source and the senor for measuring the spectra in the field?

AC47: "In-situ reflectance spectra were collected just before grab sampling at the corresponding location with an accessory light source (contact probe) mounted on the light-collecting head of the ASD spectroradiometer" (section 3.4.1). For measurement, the ASD contact probe is put into direct contact with the object. It contains an internal light source.

The authors mention that spectras measured in loose material should not be compared to those measured in filters (please add references in page 12-line 6-10). I wonder if the authors have consider transferring the loose materials into filters, to measure the spectra and be able to compare the soil and sediment information.

AC48:  The requested reference will be added. Transferring the loose material to filters could indeed be done. However, as the specific campaign aimed at collecting spectral information to be used in remote sensing, this was not meaningful in that context and not performed.

I would also appreciate a plot showing the measuring gaps for each data set (i.e. variable/station). I would encourage the authors to find a visual way to show the quality of the collected data in a plot.

AC49: We will prepare a plot showing data coverage.

As for data quality, we have included all data that we considered of potential use. Wherever these data is apparently still problematic (e.g. unusual values in turbidity) the respective values are tagged accordingly following CUAHSI conventions and excluded from further processing (i.e. conversion to SSC). These explanations will be added to 3.3.2.

"Quality" of discharge data is expressed by its uncertainty bounds (see 3.2.3) contained in the database. Considering the number and length of these time series, we do not see a feasible option for displaying this information.

We have no indication for assessing the quality of the other time series.

In the variables table. Are water stage and discharge data 'average' data or punctual?

AC50: As correctly stated in the table, water stage is averaged by the sensors during the measurement interval. Consequently, discharge derived thereof is also an average. However, given the relatively high resolution, the difference to a punctual measurement would be negligible.

Also, why variable name for reflectance is 'albedo'?

AC51: "Albedo" is a keyword of the controlled vocabulary of CUAHSI (http://his.cuahsi.org/mastercvreg/edit_cv11.aspx?tbl=VariableNameCV&id=1157579162) . It is defined as "The ratio of reflected to incident light.", which is identical to reflectance.

elevation data could added for the soil sites. Almost no information is provided for the soil sampling sites in the database, only land use type (e.g. 'grassland'), could addition information be added (e.g. soil type, organic matter content,...)

AC52: Elevation will be added. No other information is available for the sampling sites (see AC46).

SPECIFIC COMMENTS - You refer to the study site as 'dryland region' (e.g. Page 2-Line 5, Page 2-line23). Having in mind that the Isábena catchment has a Mediter-ranean climate, I wonder if this is correct to use this terminology. If I remember it well, other authors use terms as 'humid Mediterranean catchments with badland'. –

AC53: Sentence will be corrected.

Page 2, lines 10-11. I disagree on the fact that there has been little research ('or 'much less research') on relevant landscape components for water and sediment man-agement. Please, could you explain this better or add references.

AC54: Sentence will be removed.

- Page 2, line 18.
Aren't Nord et al. 2016 presenting sediment data (i.e. hydro-meteo-sedimentological data?)? Please, check. You then say that in the following line that you present an hydro-sedimentological dataset, what about the meteo? Please, be consistent.

AC54: Reference s and terming will be corrected.

- Page 2, lines 27-28: Just a suggestion. I would like to read here the measurement period and the catchment area.
AC55: we will add this information.

- Page 4-lines 8-9. Which data did you use to calculate these average values? Which period?
AC56: we will add these information.

- Page 4-line 10. Flow regime IS characterized or flow regimes are characterised. - Page 4.
AC57: Sentence will be corrected.

Sometimes you use the term 'mean' and others 'average'. I would suggest to be consistent.
AC58: We will perform these replacements.

- Page 4-line 12. I am confused with the term 'mean annual discharge' . Do you mean 'mean discharge' or 'mean punctual discharge' or 'mean instantaneous discharge'? which data period have you use to estimate these values?
AC59: We will remove "mean" and specify the data period.

- Page 4, last paragraph. Please, revise punctuation. –
AC60: Sentence will be corrected.

Page 5-line 21-22. Why data from 2004 to 2010 is not included in the dataset?
AC60: . While some data (Q, P) have also been collected before 2010, the respective methods (instruments, resolution, locations, protocols) differed. We therefore chose to present the consistent part of the dataset only.  (see AC8).

- Page
6, lines 3-4. Please, reformulate the sentence.
AC60: Sentence will be rephrased.